# Repeatability and Reproducibility of Retinal Fractal Dimension Measured with Swept-Source Optical Coherence Tomography Angiography in Healthy Eyes: A Proof-of-Concept Study

**DOI:** 10.3390/diagnostics12071769

**Published:** 2022-07-21

**Authors:** Louis Arnould, Déa Haddad, Florian Baudin, Pierre-Henry Gabrielle, Marc Sarossy, Alain M. Bron, Behzad Aliahmad, Catherine Creuzot-Garcher

**Affiliations:** 1Ophthalmology Department, Dijon University Hospital, 21000 Dijon, France; haddad.dea@gmail.com (D.H.); florian.baudin@chu-dijon.fr (F.B.); phgabrielle@gmail.com (P.-H.G.); alain.bron@chu-dijon.fr (A.M.B.); catherine.creuzot-garcher@chu-dijon.fr (C.C.-G.); 2INSERM, CIC1432, Clinical Investigation Center, Clinical Epidemiology/Clinical Trials Unit, Dijon University Hospital, 21000 Dijon, France; 3Taste and Food Science Centre, AgroSup Dijon, CNRS, INRAE, Bourgogne Franche-Comté University, 21000 Dijon, France; 4EA7460, PEC2, Cerebral and Cardiovascular Epidemiology and Physiopathology, 21000 Dijon, France; 5School of Engineering, RMIT University, Melbourne, VIC 3000, Australia; marc@sarossy.com (M.S.); b.aliahmad@gmail.com (B.A.)

**Keywords:** fractal dimension, healthy volunteers, swept-source OCT angiography, superficial retinal capillary plexus, deep retinal capillary plexus, retina, vascular imaging

## Abstract

The retinal vascular network fractal dimension (FD) could be a promising imaging biomarker. Our objective was to evaluate its repeatability and reproducibility in healthy eyes. A cross-sectional study was undertaken with young, healthy volunteers who had no reported cardiac risk factors or ocular disease history. For each participant, three SS-OCTA images (12 × 12 mm) were acquired using the Plex Elite 9000 (Carl Zeiss Meditec AG, Jena, Germany) by two ophthalmologists. Automated segmentation was obtained from both the superficial and deep capillary plexuses. FD was estimated by box counting. The intraclass correlation coefficients (ICC) were used as measures for repeatability and reproducibility. A total of 43 eyes of healthy volunteers were included. The mean ± standard deviation (SD) age was 30 ± 6.2 years. The results show good repeatability. The ICC was 0.722 (95% CI, 0.541–0.839) in the superficial capillary plexus and 0.828 (95% CI, 0.705–0.903) in the deep capillary plexus. For reproducibility, the ICC was 0.651 (95% CI, 0.439–0.795) and 0.363 (95% CI, 0.073–0.596) at the superficial and deep capillary plexus, respectively. In this study, the FD of the vascular network measured via SS-OCTA showed good repeatability and reproducibility in healthy participants.

## 1. Introduction

Quantitative features of the retinal vascular network may be indicative of the systemic microvascular network and may help predict future cardiovascular and cerebrovascular events [1,2,3,4,5,6]. Retinal arteriole narrowing and retinal vein widening on retinal fundus photographs were shown to be associated with increased long-term cardiovascular outcomes [7]. However, the retinal vascular network could provide more information. Advances in retinal imaging and automated image processing have shed new light on associations between more complex retinal vessel metrics such as the fractal dimension (FD) and cardiovascular and cerebrovascular disorders [8].

Mainster was the first to describe the retinal microvascular network as a fractal [9], and it can be interpreted as a geometric index of complexity. Indeed, retinal vascular FD represents a global measure of retinal vascular network complexity and density. Moreover, FD can quantify the complex arborisation distribution of the retinal microvascular network. The FD of the retinal vascular network has already been reliably measured from fundus photographs [10,11]. Previous studies based on fundus photography have suggested that the FD may be an effective method to predict the early vascular progression of diabetic retinopathy [12], to identify patients with occult proliferative diabetic retinopathy [13] and vascular systemic disorders such as cardiovascular burden. In order to measure retinal FD, retinal images need to be processed to extract the pattern of the retinal vascular tree. After the vessel segmentation step, image binarisation is performed and then fractal analysis is calculated. The retinal FD lies between 1 and 2. A lower FD could suggest a vascular rarefaction and a higher FD could attest to a healthy vascular status. Hence, FD as a global assessment of the architecture of the retinal vascular network could be a noninvasive surrogate biomarker of the systemic vascular status.

Optical coherence tomography angiography (OCTA) may improve the assessment of the retinal vascular network in daily clinical practice. OCTA is a non-invasive imaging technique for the assessment of the retinal microcirculation without the use of dye injections [14,15]. OCTA makes it possible to separately study both the superficial capillary plexus (SCP) and the deep capillary plexus (DCP), which cannot be individualised from the images acquired via standard angiography [16]. In that respect, OCTA benefited from an excellent depth resolution and volumetric data. Moreover, imaging SCP and DCP could be of paramount importance for vascular retinal disease description and quantification or research into vascular pathogenesis. Recently introduced swept-source optical coherence tomography (SS-OCT), which uses a short-cavity swept laser with a tuneable wavelength, offers higher imaging speeds, greater detection efficiency, an improved imaging range, and deeper penetration with reduced sensitivity roll-off [17] compared with spectral domain OCT (SD-OCT). Currently, it is possible to automatically assess vascular density, vascular perfusion, and the foveal avascular zone area by means of SS-OCTA. Previous studies reported high reproducibility for vascular density and the foveal avascular zone area with OCTA in healthy volunteers and in patients with retinal diseases [18,19,20]. Up until now, measuring the retinal vascular network FD with SS-OCTA remains complex and there is no normative database available [21]. We decided to investigate FD because it can potentially quantify the retinal microvascular network in a single scalar quantity. Retinal vascular network FD has been widely studied with fundus photography [9,10,12] and SD-OCTA [22,23], but the literature on FD and SS-OCTA is sparse. Moreover, the merit of this non-invasive parameter has been proven in various multidisciplinary areas combining ophthalmology with cardiology [24,25], neurology [8,26], and endocrinology [12,27]. Nevertheless, before using this metric in clinical practice or clinical trials, it is essential to determine the repeatability and reproducibility of the measurement. Repeatability is defined as the agreement of measurements obtained from different scanning sessions with the same device, the same eye and the same operator. In contrast, reproducibility refers to the agreement between different operators with the same eye [28].

In this study, we evaluate the utility of the retinal vascular network FD, by assessing its baseline values, and assess the repeatability and reproducibility of this measure when calculated from SS-OCTA images.

## 2. Materials and Methods

### 2.1. Study Participants

We conducted a cross-sectional descriptive study in the department of ophthalmology at Dijon University Hospital, France, between 1 November 2017, and 1 June 2018. The protocol was approved by the regional ethics committee and followed the tenets of the Declaration of Helsinki. All patients gave their written informed consent for the examinations before enrolment. We followed the GRRAS (Guidelines for Reporting Reliability and Agreement Studies) [29].

We enrolled consecutive healthy volunteers between 18 and 55 years of age. The participants’ demographic data were collected, including age, sex, and ocular history. Participants with a self-reported eye disease, refractive abnormalities (high myopia < −6 diopters, hyperopia > +6 diopters, astigmatism < −3 diopters), cardiovascular history or neurological history were excluded from the analysis. All participants benefited from a comprehensive eye examination and SS-OCTA acquisition on the same day. We also measured blood pressure with an automated device, after 5 min of rest in a seated position. Weight and height were recorded to calculate each participant’s body mass index (BMI). Globe axial length was measured using the IOL master 500 (Carl Zeiss Meditec, Jena, Germany). Measurements were taken on both eyes, but the data from one eye only were retained for analysis, according to the following procedure: (1) SS-OCTA of the right eye for participants born in even-numbered years and the left eye for those born in odd-numbered years; (2) if a scan was uninterpretable for one eye, the scan of the other eye was retained for analysis; and (3) if both eyes had inadequate signal strength (<7/10) or abnormal segmentation or significant artefacts on the perfusion map (lines or gap), the participant was excluded.

### 2.2. Image Acquisition and Processing

SS-OCTA images were acquired with the Plex Elite 9000 v1.5 (Carl Zeiss Meditec, Jena, Germany), which uses a swept laser source with a central wavelength between 1040 nm and 1060 nm and a rate of 100,000 A-scans per second. All participants underwent SS-OCTA examination under mydriasis obtained with tropicamide 0.5% eyedrops (Thea, Clermont-Ferrand, France). The study participants underwent imaging in a single visit, including three sets of scans. Two independent, trained observers performed the scans of the 12 × 12 mm region. Three 12 × 12 mm scans were performed for each participant: the first by observer 1, the second and the third by observer 2. In this way, the repeatability analysis was assessed with scans 2 and 3 performed by observer 2, and the reproducibility analysis was measured with scan 1 (observer 1) and scan 2 (observer 2). We investigated both superficial and deep retinal capillaries (Figure 1), which were automatically segmented by the SS-OCTA system software. Automatic projection artefact removal algorithm was not applied since it was not available on our SS-OCTA device at the time of study inclusion. The superficial retinal capillary layer was defined as the layer which starts from the internal limiting membrane (ILM) and extends to the inner plexiform layer (IPL). The deep retinal capillary layer was defined as the layer which starts from the IPL and extends to the outer plexiform layer (OPL).

### 2.3. Retinal Vascular Network Fractal Dimension Analysis

Fractal dimension (FD) was estimated using the binary box-counting dimension from the SS-OCTA images [30]. We used the box-counting method in FracLab for FD analysis but implemented our own pre- and post-processing routines in MATLAB. FracLab is a free software developed by the Anja team at Inria Rennes/Laboratoire Jean Leray—Université de Nantes. It is an open-source software mainly based in Matlab. Detailed information about the FD algorithm is available on the developers’ website (https://project.inria.fr/fraclab/ (accessed on 1 January 2019) and the code is freely available to the public.

For this method, images were first binarised. This is a process where the greyscale intensity of all pixels in the image were converted to either 1 or 0. A fixed threshold was applied to classify all pixels. This was carried out to obtain microvasculature structures in the highest possible contrast to their background, where vessels were shown in white against a black background. The next step involved image skeletonisation and successive thinning of the vasculature until each vessel was presented by its centre line with a thickness of only one pixel. All images were examined to ensure no discontinuities were introduced to the vessel structure due to the binarisation and skeletonisation process.

To obtain the box-counting FD, the skeletonised images were superimposed on a grid of squares and the total number of boxes (squares) containing at least one pixel of the structure were counted. This process was repeated continuously by reducing the mesh size in each step (increasing the total number of meshes covering the entire image). All data points (total number of meshes counted in each step vs. mesh size) were plotted on a log–log plot and the slope of the line with the best fit to the data was taken as the measurement of box-counting FD. This measure was bounded on the interval [0, 1]. Lower FD values correspond to low complexity of vessel branching, while higher values correspond to greater complexity [10]. The framework of the image processing steps is shown in Figure 2. Examples of swept-source optical coherence tomography angiography: original and processed OCTA images from a single patient as shown in Figure 3, which enables better visualisation and comparison of two processed images, and a skeleton image with modified colour to red and with superimposed original grayscale (Appendix A).

### 2.4. Statistical Analysis

Continuous variables were tested for normality (Shapiro–Wilk test). Continuous variables that follow a normal distribution are expressed as a mean ± standard deviation (range) and those that follow a non-normal distribution are expressed as a median (interquartile range). Categorical variables are summarised in numbers and percentages (*n*, %). The Student’s paired *t*-test was used to compare the differences between the scans by the two different operators.

To analyse the repeatability and reproducibility of the retinal vascular network FD, we calculated the intraclass correlation coefficient (ICC) with 95% confidence intervals (CIs) using Shrout–Fleiss reliability notation. We followed Cicchetti’s guidelines, which state that when the reliability coefficient is below 0.40, the level of clinical significance is poor; when it is between 0.40 and 0.59, the level of clinical significance is fair; when it is between 0.60 and 0.74, the level of clinical significance is good; and when it is between 0.75 and 1.00, the level of clinical significance is excellent [31]. Repeatability and reproducibility were also evaluated with the coefficient of variation (CV) calculated as the standard deviation divided by the average of the measurements multiplied by 100. A CV of less than 5% indicates good reproducibility. In addition, intra-observer repeatability and inter-observer reproducibility were evaluated by means of Bland–Altman analysis [32]. A *p*-value of less than 0.05 was considered statistically significant. All statistical analyses were performed using the statistical software SAS (version 9.4; SAS Institute, Inc., Cary, NC, USA).

## 3. Results

Of the 45 participants, 43 were included in the analysis with a mean age of 30 ± 6.2 years (range, 23–52). A total of two scans of two participants were removed due to significant artefacts on the perfusion map. Table 1 summarises the participants’ baseline demographics and clinical characteristics. None of the participants demonstrated any eye disease on clinical examination or SS-OCTA, or any pathological axial length.

The mean retinal vascular network FD in the superficial vascular layer was 1.694 ± 0.010 (range, 1.656–1.709) for observer 1 and 1.693 ± 0.014 (range, 1.610–1.708) for observer 2. The mean retinal vascular network FD in the deep vascular layer was 1.695 ± 0.007 (range, 1.662–1.704) and 1.693 ± 0.011 (range, 1.624–1.705), for observers 1 and 2, respectively (Table 2). No significant difference was found between the two observers in terms of mean retinal vascular network FD for all measurements. Bland–Altman plots were used to demonstrate differences in overall mean FD measurements for the intra-observer repeatability and inter-observer reproducibility in the SCP and DCP (Figure 4 and Figure 5). The CV for each observer was less than 5%, which indicates low variability and, therefore, good reproducibility (Table 2).

The intra-class correlation coefficients in the SCP and DCP were as follows. Intra-observer repeatability was 0.722 (95% CI, 0.541–0.839) in the SCP and 0.828 (95% CI, 0.705–0.903) in the DCP, deemed good and excellent, respectively [30]. Inter-observer reproducibility was 0.651 (95% CI, 0.439–0.795) and 0.363 (95% CI, 0.073–0.596) in the SCP and DCP, deemed good and poor, respectively, according to Cicchetti [31].

## 4. Discussion

This study aimed to describe the FD metrics of the retinal vascular network using SS-OCTCA in healthy participants, and to assess its repeatability and reproducibility in clinical practice. We used an SS-OCTA device with automatic layer segmentation to achieve 12 × 12 mm images and then quantified the retinal vascular network FD of the SCP and DCP using independent software. To our knowledge, this was the first study to investigate FD with an SS-OCTA device and wide-field images (12 × 12 mm). The mean retinal vascular network FD was 1.693 in the superficial and 1.694 in the deep vascular layer in our healthy cohort. These results are in agreement with previous studies based on fundus photographs, which found a normal retinal vessel FD of 1.7 [9,12]. Although we did not use the same imaging device, our results are consistent since we acquired wide-field 12 × 12 mm images with a field of view of 48°, which allowed us to work on the fundus area close to that obtained with fundus photographs. Previous studies have investigated retinal vessel FD using SS-OCTA. Corvi et al. evaluated the retinal vascular network FD with seven different OCTA devices including the Plex Elite 9000 (Carl Zeiss Meditec, Jena, Germany) using a 3 × 3 mm scan, and found a mean value of 1.68 and 1.69 in the superficial and deep vascular plexuses, respectively [18]. Although the acquisition size in our study was different, our results are in agreement. Hirano et al. quantified the retinal vascular network FD with the same SS-OCTA device and image size in healthy participants and in patients with diabetes [33]. They found a mean retinal vascular network FD of 1.627 and 1.612 in the superficial and deep capillary network in healthy eyes, respectively. However, to our knowledge, this paper is the first to assess the repeatability and reproducibility of this type of FD. Furthermore, it should be noted that retinal vascular network FDs in the SCP and DCP were fairly similar. Few studies have compared the retinal vascular network FD in these two networks. It has previously been reported that the patterns of these two networks are different, and that the central avascular zone is larger in the deep network compared with the superficial one [34]. FD evaluates the retinal vascular network as a whole, which may explain why there were only a few differences in the mean FD between these two networks.

On the other hand, Y Lu et al. compared four different OCT-A platforms (Angiovue, Spectralis, Plex Elite, and Topcon) in 16 healthy eyes [35]. They found, using the projection artefact removal algorithm, an FD with the Plex Elite of 1.741 ± 0.060 in the SCP and 1.748 ± 0.048 in the DCP. Differences with our results could be explained by the size of SS-OCTA images (3 × 3 mm versus 12 × 12 mm) and the absence of artefact removal in our study.

The retinal vascular network FD measurements showed good repeatability and reproducibility in our study. The measurements obtained by the same operator and those obtained by two operators were not statistically significantly different when the same protocol was used. Intra-observer repeatability was good in the SCP and excellent in the DCP. However, we found that the measurements of the retinal vascular network FD in the DCP did not have very good inter-observer reproducibility. This could be explained by the difficulty of removing the projection artefacts of vessels from the superficial vascular network to the deep vascular network. It has already been shown that analysis of the superficial retinal plexus is more accurate than the deeper plexus because of projection artefacts [2]. Another explanation could be the motion artefacts or modification of the centration of the image between two OCTA acquisitions. However, this did not occur between two acquisitions made by the same observer and, therefore, it could be related to changes in the head position between the two observers. This could explain the surprisingly great differences between ICC for the reproducibility of deep plexus (0.363) when the repeatability is 0.828. Even small head movements, eye misposition or differences in the patient’s head position can produce dramatic changes from one scan to another, although the SS-OCTA device eye-tracker tried to control for these artefacts. Moreover, peripheral shadowing could also explain the difference in FD results when comparing several SS-OCTA exams. These parameters highlighted the challenges encountered to compare FD measurements from different patients and examiners.

Even though previous studies focused on the repeatability of quantitative metrics other than FD using widefield SS-OCTA, we can compare our results to them [36,37]. J Hong et al. investigated the inter-visit repeatability of 15 × 9 mm^2^ SS-OCTA and they demonstrated highly repeatable measurements of retinal vascular metrics [36]. They also showed that ICC values in the DCP were generally lower than those of the SCP. J Hong et al. reported lower ICC in the DCP (0.72) compared to the SCP (0.84) for vessel density measurement using 12 × 12 mm SS-OCTA [37]. These differences between SCP and DCP could be also explained by the fact that SCP has larger capillaries compared to the finer capillaries in the DCP. The same rationale may apply to the retinal FD measurement.

The Bland–Altman plot of the intra-observer repeatability of the retinal vascular network fractal dimension in the SCP showed that differences increased significantly with decreasing FD measurements (Figure 5). This could introduce a proportional bias as lower FD could suffer from a higher level of noise.

The results regarding reproducibility in this study are only applicable to the SS-OCTA Plex Elite 9000 (Carl Zeiss Meditec, Jena, Germany). Corvi et al. compared retinal vascular network FD measurements between seven different OCTA devices and found that the mean retinal vascular network FD in both the SCP and DCP was different between these instruments [18]. Differences were even found between devices using the same algorithm and segmentation limits. Thus, measurements between different OCTA devices were not interchangeable. Moreover, we only included healthy eyes in our study and, therefore, the automated segmentation is more accurate than, for example, in the presence of haemorrhage. We should expect that in eyes with retinal disorders, automatic segmentation might decrease the repeatability of these measurements.

In previous studies, the retinal vascular network FD was considered a promising biomarker for the study of retinal and cardiovascular diseases. These studies used methods based on the measurement of FD from fundus photographs [12,27]. With fundus photographs, it is impossible to distinguish the two vascular networks. Current advances in retinal imaging have allowed us to describe the retinal vascular network thoroughly. Furthermore, the swept-source technique has the advantage of being able to acquire larger images with better quality and contrast than SD-OCT, thus allowing us to study the vascular network as a whole [17]. Moreover, the benefit of using a 12 × 12 mm wide-field acquisition zone is the ability to simultaneously visualise the macula region and the optic nerve head, and to analyse both the macular network and radial peripapillary capillaries on the superficial and deep vascular plexus.

Fractal analysis can provide greater insight into the development of retinal vascular diseases. Recent studies have demonstrated the value of the retinal vascular network FD in clinical practice. A reduced retinal vascular network FD was found in all stages of diabetic retinopathy with SS-OCTA [33,38]. Nevertheless, retinal vascular network FD analysis of the retinal network is a global measure of the blood vessel pattern; as such, it is not sensitive to minor focal alterations [10]. An earlier study explored the relationship between the retinal vascular network FD and neovascular age-related macular degeneration (nAMD) [39]. The relationship between quantitative OCTA parameters in patients with active nAMD under treatment and those with remission nAMD was investigated. The retinal vascular network FD in the active nAMD group was significantly lower than that in the remission group (1.44 vs. 1.50, *p* < 0.001) [39]. This was explained by the increased branching after arteriogenesis in the remission group. The retinal vascular network FD therefore has potential for the study of retinal vascular diseases and macular disorders. We decided to evaluate FD in wide-field 12 × 12 mm images (and not 3 × 3 or 6 × 6 mm) in order to capture more thorough and broader vascular information. We hypothesised that 12 × 12 mm FD could be a promising biomarker of the retinal vascular status and a potential surrogate to cerebro- and cardiovascular status. Hence, in future studies it would be interesting to evaluate its contribution to the diagnosis of cerebro- and cardiovascular disorders such as stroke and myocardial infarction. Combined with other quantitative parameters such as vascular density, vascular perfusion, and the area of nonperfusion, the retinal vascular network FD offers the possibility of an additional way to monitor retinal disease and to refine cardiovascular risk stratification.

There were some limitations of this study that should be mentioned. First, it was a small-sample single-centre study with a single SS-OCTA device, which may limit its external validity. Secondly, it was a self-report study; we did not carry out a medical assessment or blood test to eliminate vascular disease such as diabetes. Thirdly, the age range was limited to individuals aged 24–52 years. Fourthly, we had artefacts during OCTA image acquisition, including motion artefacts and projection artefacts [40]. In this setting, FD measurement and analysis in the DCP were challenging. Future work includes recalculation with the automatic projection artefact removal algorithms that are now available with the new SS-OCTA, which would strengthen these findings.

## 5. Conclusions

In conclusion, the retinal vascular network FD provided new repeatable and reproducible quantitative data using SS-OCTA with healthy participants. The clinical relevance of these findings warrants further studies.

## Figures and Tables

**Figure 1 diagnostics-12-01769-f001:**
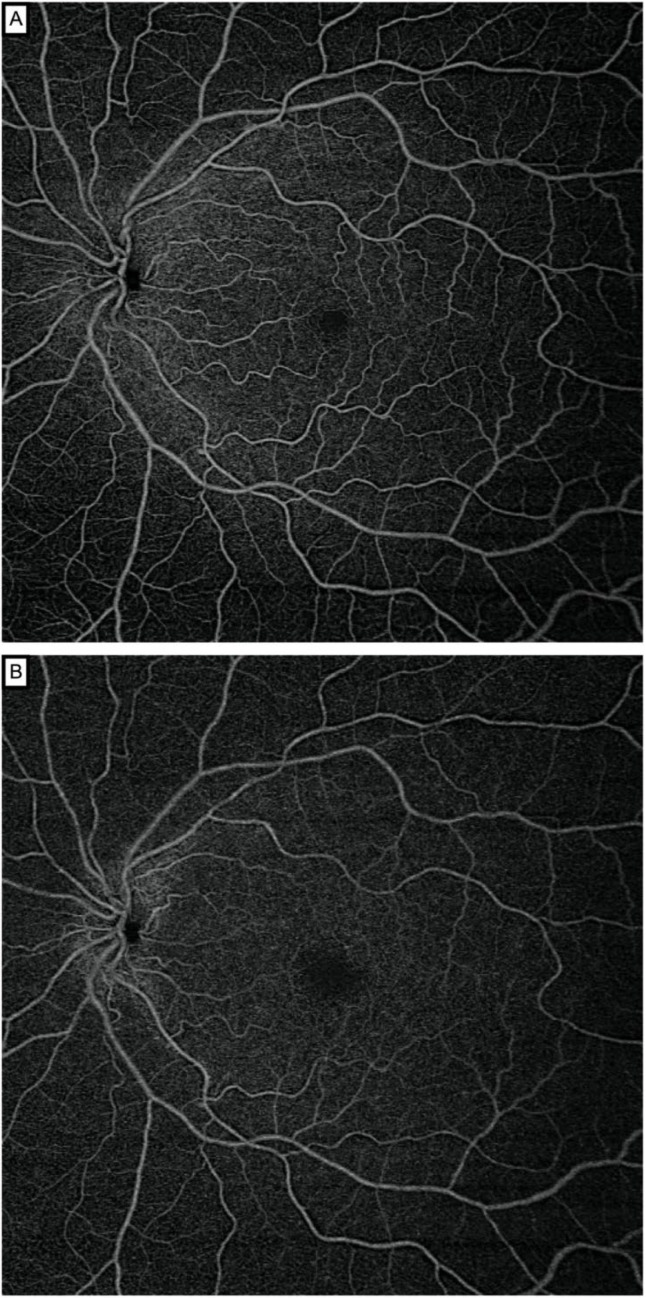
Swept-source optical coherence tomography angiography 12 × 12 mm en face images segmented through the superficial (**A**) and deep capillary plexus (**B**) of a healthy subject. The Plex Elite automatically provided an image of the SCP and DCP, using default slabs, the inner and outer boundaries of which were set at the inner limiting membrane and inner plexiform layer–inner nuclear layer (SCP) interface and the inner plexiform layer–inner nuclear layer and outer plexiform layer–outer nuclear layer (DCP) interfaces.

**Figure 2 diagnostics-12-01769-f002:**
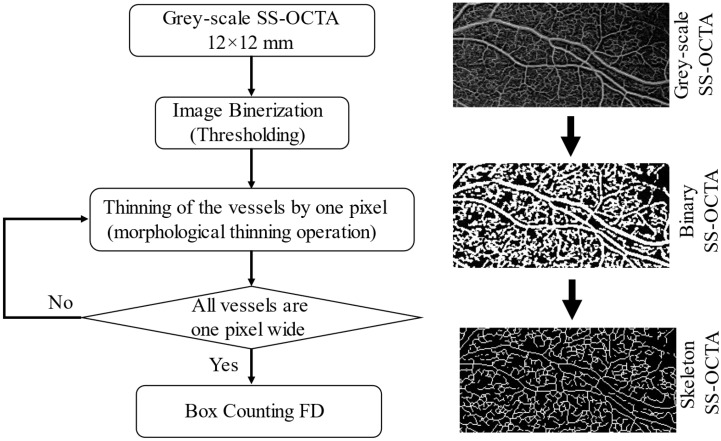
Framework of image processing steps.

**Figure 3 diagnostics-12-01769-f003:**
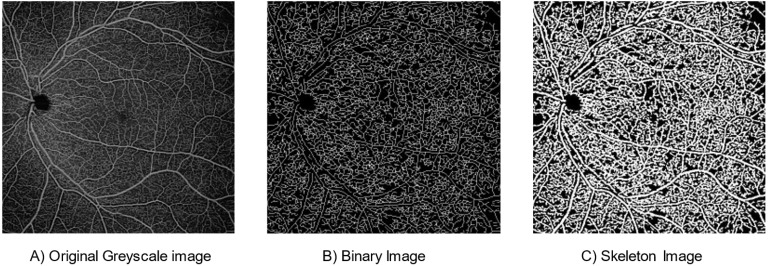
Swept-source optical coherence tomography angiography: Original and processed OCTA images from a single patient.

**Figure 4 diagnostics-12-01769-f004:**
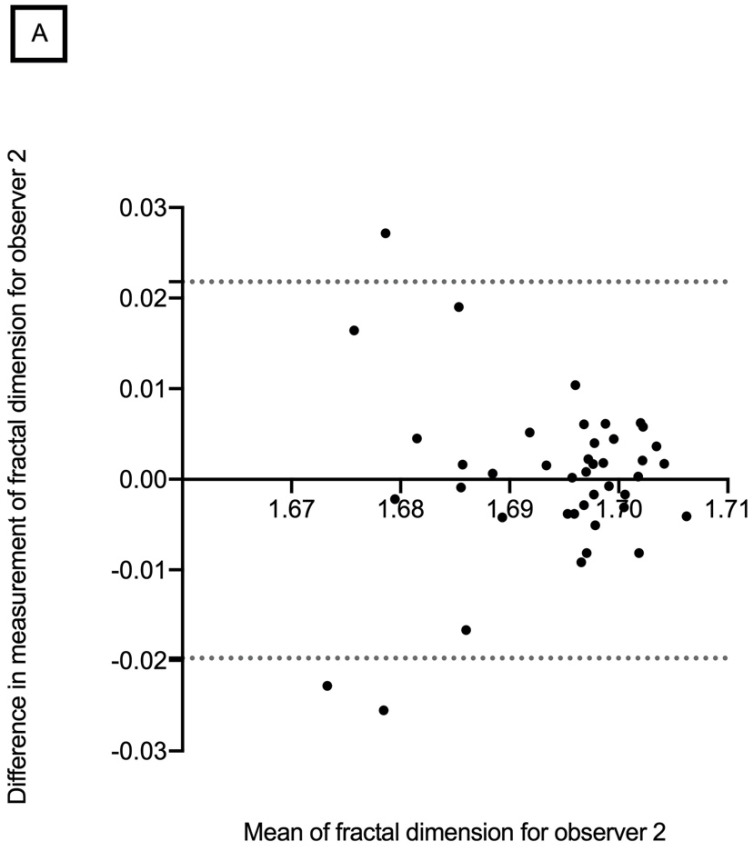
Bland–Altman plots showing intra-observer repeatability of retinal vascular network fractal dimension in the superficial (**A**) and deep capillary plexus (**B**).

**Figure 5 diagnostics-12-01769-f005:**
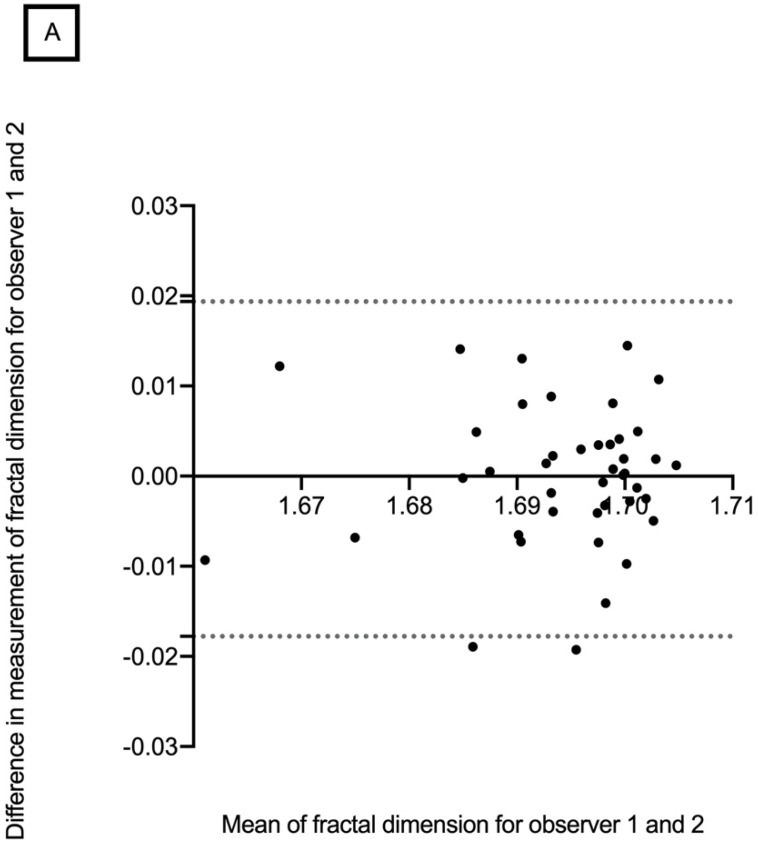
Bland–Altman plots showing inter-observer reproducibility of retinal vascular network fractal dimension in the superficial (**A**) and deep capillary plexus (**B**).

**Table 1 diagnostics-12-01769-t001:** Baseline characteristics of the participants.

Baseline Characteristics	
Age, years	30 ± 6.2
Sex, female	24 (55.82)
Study eye, right	29 (67.44)
Axial length, mm	23.7 ± 1.4
Body mass index, kg/m^2^	21.8 ± 2.1
Systolic blood pressure, mmHg	128.9 ± 7.9
Diastolic blood pressure, mmHg	79.3 ± 10.2

Continuous variables are displayed as mean ± standard deviation. Categorical variables are displayed as number (percentage).

**Table 2 diagnostics-12-01769-t002:** Mean fractal dimension for the two observers.

	Observer 1	Observer 2	*p*-Value
Average fractal dimension			
SCP	1.694 ± 0.010	1.693 ± 0.014	0.59
DCP	1.695 ± 0.007	1.693 ± 0.011	0.41
Coefficient of variation			
SCP	0.60	0.43	
DCP	0.83	0.64	

Continuous variables are displayed as mean ± standard deviation. Coefficient of variation are displayed as percentage. SCP = superficial capillary plexus; DCP = deep capillary plexus.

## Data Availability

Data are available on reasonable request. A data request form has to be sent to louis.arnould@chu-dijon.fr.

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
