# Peer review of "Repeatability and Reproducibility of Retinal Fractal Dimension Measured with Swept-Source Optical Coherence Tomography Angiography in Healthy Eyes: A Proof-of-Concept Study"

_diagnostics, 2022, doi:10.3390/diagnostics12071769_

Round 1

Reviewer 1 Report

  • This manuscript conducts a baseline measurement of fractal dimension based on SS-OCT images of healthy eyes. The manuscript has been improved significantly since last submission. Here are my new comments regarding the second version:
  • Where is Figure 5?
  • Your first sentence of the paper: "Retinal vascular network fractal dimension (FD) could be an interesting imaging biomarker." --- you probably want to replace "interesting" by "promising" or some word similar.  
  • For the Introduction: Fractal Dimension should be separately introduced in terms of what it measures (or how it is calculated) and why this metric can be useful or what kind of feature it may represent from the images.
  • The authors discussed the disparities of their results from previous studies in the end of the paper. I wonder what would happen if the authors perform the analysis again with the automatic projection artifact removal algorithm applied. That way, you would have two sets of baseline results ready for any previous or future comparison. I believe this addition of data will make the paper much stronger. Otherwise, the authors should explain why they decided not to apply the algorithm.

Author Response

Manuscript ID: diagnostics-1787867

Type of manuscript: Article
Title: Repeatability and reproducibility of a quantitative assessment of the retinal vascular network fractal dimension measured with swept-source optical coherence tomography angiography in healthy eyes: a proof-of-concept

Thank you for giving us the opportunity to revise this submission following this third peer review. After consultation with the other authors, we have responded to the editor’s and reviewers’ constructive suggestions by revising the manuscript as attached. We would like to take the opportunity to respond to the individual points below.

Reviewer 1

This manuscript conducts a baseline measurement of fractal dimension based on SS-OCT images of healthy eyes. The manuscript has been improved significantly since last submission. Here are my new comments regarding the second version:

Where is Figure 5?

Figure 5 could be found in page 26 of the manuscript just before Table 1.

It is also presented below.

Fig. 5 Bland–Altman plots showing inter-observer reproducibility of retinal vascular network fractal dimension in the superficial  (A) and deep capillary plexus (B)

Your first sentence of the paper: "Retinal vascular network fractal dimension (FD) could be an interesting imaging biomarker." --- you probably want to replace "interesting" by "promising" or some word similar.  

We thank the Reviewer for this note. As suggested, we modified the revised version of the manuscript.

Abstract

Retinal vascular network fractal dimension (FD) could be a promising imaging biomarker.”

Line 273-274

“In previous studies, the retinal vascular network FD was considered a promising biomarker for the study of retinal and cardiovascular diseases.”

Line 298-300

“We hypothesized that 12X12 mm FD could be a promising biomarker of the retinal vascular status and a potential surrogate to cerebro and cardiovascular status.”

For the Introduction: Fractal Dimension should be separately introduced in terms of what it measures (or how it is calculated) and why this metric can be useful or what kind of feature it may represent from the images.

As suggested by the Reviewer, we modified as follow the Introduction. Moreover, we strongly believe that retinal FD metric is a promising biomarker because it could be measured independently of the OCT-A device and the manufacturer’s algorithm which could help generalization of research findings.

Line 50-64:

“Mainster was the first to describe the retinal microvascular network as a fractal [9] and it can be interpreted as a geometric index of complexity. Indeed, retinal vascular FD represents a global measure of retinal vascular network complexity and density. Moreover FD can quantify the complex arborization distribution of the retinal microvascular network. The FD of the retinal vascular network has already been reliably measured from fundus photographs [10, 11]. Previous studies based on fundus photography have suggested that the FD may be an effective method to predict early vascular progression of diabetic retinopathy [12], to identify patients with occult proliferative diabetic retinopathy [13] and vascular systemic disorder such as cardiovascular burden. In order to measure retinal FD, retinal images need to be processed to extract pattern of the retinal vascular tree. After the vessel segmentation step, image binarization is performed and then fractal analysis is calculated. The retinal FD lies between 1 and 2. A lower FD could witness a vascular rarefaction and a higher FD could attest of a healthy vascular status. Hence, FD as a global assessment of the architecture of the retinal vascular network could be a noninvasive surrogate biomarker of the systemic vascular status.”

The authors discussed the disparities of their results from previous studies in the end of the paper. I wonder what would happen if the authors perform the analysis again with the automatic projection artifact removal algorithm applied. That way, you would have two sets of baseline results ready for any previous or future comparison. I believe this addition of data will make the paper much stronger. Otherwise, the authors should explain why they decided not to apply the algorithm.

We want to thank the Reviewer for emphasizing this crucial point. Additional data sets is not possible for this present study. We modified the Methods and Discussion sections as follow

Line 131-132

“Automatic projection artefact removal algorithm was not applied since it was not available on our SS-OCTA device at the time of study inclusion.”

Line 320-322

“Future work includes recalculation with the automatic projection artefact removal algorithms that are now available with new SS-OCTA would strengthen our findings.”  

Reviewer 2 Report

The authors of this paper evaluate the utility of the retinal vascular network fractal dimension (FD), by assessing its baseline values, and assess the repeatability and reproducibility of this measure when calculated from Swept-Source Optical coherence tomography (SS-OCTA) angiography images.

The subject of the paper is quite interesting and very well presented. Some minor syntax/grammar errors can be corrected during the preparation of the camera ready version. Based on the above the reviewer believes that the paper can be accepted for publication.

Author Response

Manuscript ID: diagnostics-1787867

Type of manuscript: Article
Title: Repeatability and reproducibility of a quantitative assessment of the retinal vascular network fractal dimension measured with swept-source optical coherence tomography angiography in healthy eyes: a proof-of-concept

Thank you for giving us the opportunity to revise this submission following this third peer review. After consultation with the other authors, we have responded to the editor’s and reviewers’ constructive suggestions by revising the manuscript as attached. We would like to take the opportunity to respond to the individual points below.

Reviewer 2

The authors of this paper evaluate the utility of the retinal vascular network fractal dimension (FD), by assessing its baseline values, and assess the repeatability and reproducibility of this measure when calculated from Swept-Source Optical coherence tomography (SS-OCTA) angiography images.

The subject of the paper is quite interesting and very well presented. Some minor syntax/grammar errors can be corrected during the preparation of the camera ready version. Based on the above the reviewer believes that the paper can be accepted for publication.

We want to thank the Reviewer for their very positive insights. As suggested, the manuscript was double checked by native English co-authors (MS, BA) for syntax/grammar errors.
